# Compromised Mitotic Fidelity in Human Pluripotent Stem Cells

**DOI:** 10.3390/ijms241511933

**Published:** 2023-07-25

**Authors:** Inês Milagre, Carolina Pereira, Raquel A. Oliveira

**Affiliations:** 1Católica Biomedical Research Centre, Católica Medical School, Universidade Católica Portuguesa, 1649-023 Lisbon, Portugal; 2Instituto Gulbenkian de Ciência, 2780-156 Oeiras, Portugal

**Keywords:** human pluripotent stem cells, mitotic fidelity, aneuploidy

## Abstract

Human pluripotent stem cells (PSCs), which include both embryonic and induced pluripotent stem cells, are widely used in fundamental and applied biomedical research. They have been instrumental for better understanding development and cell differentiation processes, disease origin and progression and can aid in the discovery of new drugs. PSCs also hold great potential in regenerative medicine to treat or diminish the effects of certain debilitating diseases, such as degenerative disorders. However, some concerns have recently been raised over their safety for use in regenerative medicine. One of the major concerns is the fact that PSCs are prone to errors in passing the correct number of chromosomes to daughter cells, resulting in aneuploid cells. Aneuploidy, characterised by an imbalance in chromosome number, elicits the upregulation of different stress pathways that are deleterious to cell homeostasis, impair proper embryo development and potentiate cancer development. In this review, we will summarize known molecular mechanisms recently revealed to impair mitotic fidelity in human PSCs and the consequences of the decreased mitotic fidelity of these cells. We will finish with speculative views on how the physiological characteristics of PSCs can affect the mitotic machinery and how their suboptimal mitotic fidelity may be circumvented.

## 1. Human Pluripotent Stem Cells Present Frequent Karyotypic Abnormalities

Pluripotent stem cells (PSCs) can give rise to all cell types of the organism [1]. Their potential for self-renewal and differentiation makes them very appealing for use in basic research as well as drug screening and discovery, toxicity tests and ultimately in cellular therapy [2]. Human embryonic stem cells (ESCs), a type of PSC, were first derived from human embryos in 1998 [3]. These cells have the capacity for self-renewal, can be cultured in vitro indefinitely and have the ability to differentiate into cells of the three germ layers: endoderm, ectoderm and mesoderm [3]. In 2006, the revolutionary work by Takahashi and Yamanaka demonstrated that it was possible to reprogram adult and embryonic fibroblasts to obtain “embryonic stem cell-like” cells, which they termed induced pluripotent stem cells (iPSCs) [1], another type of PSC. By induced expression of only four transcription factors (Oct4, Sox2, Klf4 and c-Myc), mouse fibroblasts were reprogrammed into iPSCs. Only a year later, Yamanaka and others showed it was also possible to use the same minimal cocktail of proteins to obtain iPSCs from several human cell types [4,5,6]. Already numerous clinical trials are ongoing using either ESCs or iPSCs, aiming to cure and/or ameliorate symptoms of a plethora of human diseases, including many degenerative disorders and ageing-related complications, such as Parkinson’s disease, macular degeneration and heart failure [7]. It is therefore essential that these cells are safe and robust while maintaining a stable karyotype (the species’ complete set of chromosomes), since karyotypic abnormalities hinder the safe use of PSCs in regenerative medicine and in translational and basic research [7,8]. Aneuploidy—an abnormal chromosome number—can impair proper embryo development, potentiate cancer development and trigger different types of stresses in cells [9]. Yet, PSCs are sometimes aneuploid, frequently having partial or whole chromosome gains or losses [10,11,12,13] and/or recurrent copy number variations [14,15,16].

Aneuploidy occurs in a wide range of biological settings. For example, mammalian female meiosis is highly erroneous, resulting in aneuploid embryos [17]. Additionally, early embryos accumulate aneuploid cells due to mitotic errors [18]. Aneuploidy is also a key aspect of tumour biology, found in around 90% of solid tumours and 75% of hematopoietic cancers [19,20,21]. Likewise, culture conditions, such as medium acidification, may affect the genomic stability of PSCs [22]: up to 30% of PSC lines used in laboratories all over the world present karyotypic abnormalities, including whole chromosome aneuploidies [10,11]. Yet, we are only now starting to characterize and understand the molecular mechanisms underlying this chromosomal imbalance, its link to pluripotency and the consequences for PSC maintenance. Recently, several papers have begun to clarify the specific mitotic pathways that are malfunctioning in human PSCs. In this review we will focus on the known molecular pathways and physiological properties that can contribute to whole chromosome gains or losses in human PSCs. We will start by summarising recent findings that have identified specific mitotic molecular pathways that are malfunctioning or suboptimal in PSCs. Next, we will discuss how certain properties of these cells may underlie their reduced mitotic fidelity and how particular differences in the cellular responses to abnormal chromosome numbers may contribute to a differential survival of aneuploid PSCs. We will finalize with speculative views on how basic knowledge on the properties that render PSCs more prone to mitotic defects can be used to improve their mitotic fidelity and, thus, their safe applicability.

## 2. Molecular Mechanisms Underlying the Compromised Mitotic Fidelity in Pluripotent Stem Cells

PSCs have been described as having karyotypic abnormalities for more than a decade [10,11,12,13]; however, the molecular mechanisms that account for the problems in mitosis have only recently been explored (see Figure 1 and Table 1).

Proper chromosome segregation is central to genome stability and the maintenance of the correct karyotype relies on the mitotic machinery that segregates chromosomes with high fidelity [32,33,34,35,36,37]. PSCs have an atypical cell cycle structure with truncated gap phases and proliferate at an unusually rapid rate [38]. Additionally, the chromatin of PSCs is characterised by being open, decondensed and more dynamic than that of differentiated cells [39]. It is possible that the intrinsic characteristics of the cell cycle and chromatin state impose differences in mitotic machinery function and regulation in PSCs, resulting in compromised mitotic fidelity.

### 2.1. DNA Replication

Several studies have shown that PSCs undergo much higher replication stress during S phase [23,27,28,29,30] than differentiated cells, although the duration of S phase is comparable in both cell types. Replication stress, such as incomplete, under-replicated or unresolved chromosomes from S phase, if not corrected, can persist into mitosis and hinder proper chromosome segregation [40]. Replication stress in PSCs arises mainly due to slower rates of DNA synthesis, activation of latent origins of replication and stalling of replication forks, which results in abortive and abnormal mitosis [23,27]. The high levels of replication stress that these cells undergo result in DNA damage and double strand breaks [27]. It is likely that these problems during replication are a consequence of the truncated cell cycle, characterised by a very fast G1 [38], which would preclude cells from properly preparing for the subsequent replication of DNA in the S phase. In accordance with this, the addition of exogenous nucleosides to the culture medium of PSCs results in a partial rescue of the replication dynamics, reducing the levels of DNA damage [27,41], decreasing the incidence of abnormal mitosis and increasing survival of these cells [27].

The response to replication stress of PSCs is distinct from that of differentiated cells and may result in the increased aneuploidy of these cells. In response to this type of stress, PSCs tend to fail to activate DNA repair pathways and favour instead the activation of apoptosis [42], which may help control the deleterious effects of replication stress. One exception to this is when telomere shortening is induced in these cells, which, combined with p53 deletion, leads to an accumulation of mitotic errors [43]. Although apoptosis is PSCs’ preferred response to DNA damage [42], they can proceed to mitosis after replication stress [23,30]. This can result in either abortive or erroneous mitosis, which can then result in daughter cells that are aneuploid. Aneuploid PSCs with an extra copy of chromosome 12, chromosome 17 or a combination of both, suffer even higher levels of replication stress and make more mistakes in mitosis, when compared to euploid PSCs [23], which could lead to further chromosomal instability.

Replication stress in PSCs has been well established [23,27,28,29,30] and has recently been shown to affect downstream mitosis [23,27,30]. However, recent papers have demonstrated that there are mitotic-specific processes that are also affected and may work in a sub-optimal way in these cells. Below we will focus on the recent advances in the mitotic processes of pluripotent stem cells and how they differ from those of differentiated cells.

### 2.2. Chromosome Condensation

In interphase, PSC chromatin is more open, mobile and accessible [39,44] than that of differentiated cells and extensive remodelling of the chromatin during iPSC reprogramming is necessary to induce pluripotency [45]. It is conceivable that these differences in mitotic chromatin properties affect how the mitotic machinery functions and is regulated in these cells, which can impact efficient chromosome segregation. During mitosis, euploid PSCs have problems in chromosome condensation, and this is exacerbated in PSCs with extra copies of chromosome 12, 17 or both [23]. These mitotic defects, as well as the increased replication stress described in the previous section of these aneuploid PSCs, seem to be due to the downregulation of a transcription factor involved in the regulation of several actin–cytoskeleton genes—the serum response factor (SRF) [23]. Overexpression of this factor in aneuploid cells resulted in a decrease in replication stress and a partial rescue of the errors and condensation defects during mitosis. Consistently, silencing SRF in euploid pluripotent stem cells led to an increase in replication stress and condensation defects [23]. However, it is unclear how known key players important for mitotic chromosome condensation in differentiated cells are regulated in either euploid or aneuploid PSCs. These include specific histone post-translational modifications [46] and key proteins, such as condensins and DNA topoisomerase II alpha [36]. Equally unclear is if differences in the regulation of these structural factors affect chromosome condensation and, therefore, the genomic integrity of pluripotent stem cells.

### 2.3. The Centromere

At the heart of faithful genome distribution is a specialised chromosomal locus—the centromere—that is present exactly once in each chromosome to ensure the binding and/or regulation of various components of the mitotic machinery. The centromere forms a scaffold for the assembly of the kinetochore, a multi-subunit protein complex that acts as a platform for microtubule attachment during mitosis to drive chromosome segregation [47]. The histone H3 variant CENP-A was identified as a crucial component in the epigenetic specification of the centromere [48]. Our own recent findings show that centromere strength is compromised in PSCs, which present reduced levels of this histone variant [24]. While PSCs maintain abundant cytoplasmatic pools of CENP-A, CENP-C and CENP-T, these essential centromere components are strongly reduced at stem cell centromeres [24]. And CENP-B, another important centromeric protein, is expressed at low levels and almost undetected at the centromeres of these cells.

The weak centromere affects the recruitment of kinetochore proteins in mitosis, at least partially, resulting in reduced levels of key proteins in kinetochore–microtubule attachment and chromosome motility [24]. Assembly of CENP-A chromatin occurs in the G1 phase of the stem cell cycle [24], as is the case in human differentiated and cancer cell lines [49,50]. The reduction in centromeric chromatin size is induced early during iPSC reprogramming [24], coincident with the time of increased proliferation rates. Additionally, chromosomes that lack CENP-B at the centromere, which is not essential for cell viability, have been shown to mis-segregate more frequently and have slightly lower levels of CENP-A and CENP-C [51]. Low CENP-A, -B and -C levels may be sufficient to impact on chromosome segregation efficiency in PSCs. Accordingly, lagging chromosomes are the major mitotic defect in these cells [25,26]. Other mitotic errors, such as anaphase bridges, micronuclei and multipolar divisions, were also detected [25,26] but did not differ greatly in frequency from those detected in differentiated cells [26]. Hence, sub-optimal CENP-A levels may be one of the most critical abnormalities in PSCs that underlies their mitotic defects.

### 2.4. Kinetochore–Microtubule Dynamics

Improper kinetochore–microtubule attachments are corrected by a dedicated error correction machinery primarily mediated by the Aurora B kinase [52]. This pathway ensures dynamic detachment and reattachment of microtubules to allow that each sister chromatid is attached to microtubules coming from one spindle pole and that the other sister is attached only to the opposite pole [53].

If error correction pathways are not working properly, cells can accumulate erroneous kinetochore–microtubule attachments at anaphase onset [54]. It has been shown that Aurora B is maintained at normal levels at the inner centromere of PSCs [24], but it is unknown if its activity is altered in these cells, which may interfere with proper error correction. Indeed, merotelic attachments (when the kinetochore of one sister chromatid is attached to one pole and the other to both poles) were observed in more than 70% of all lagging chromosomes in PSCs [26]. This high frequency is possibly the result of hyperstabilisation of microtubule attachments [26,31] concomitantly with impaired error correction. Accordingly, artificial de-stabilization of kinetochore–microtubule attachments using small molecules restores mitotic fidelity in PSCs [26]. Moreover, impairing the correct localisation of Aurora B to the inner centromere exacerbates errors in mitosis observed in PSCs [55]. These results point to altered error-correction efficiency in PSCs, which impacts on correct chromosome segregation.

### 2.5. Spindle Assembly Checkpoint

The outer kinetochore, which assembles at the centromere in mitosis, is also the site for the accumulation of proteins from the spindle assembly checkpoint (SAC). This checkpoint senses unattached or improperly attached kinetochores, preventing mitotic exit until all chromosomes are properly attached to the spindle [56]. PSCs can activate the SAC in response to spindle poisons [57], delaying anaphase onset. However, it remains unknown if SAC activation is as robust as that of differentiated cells. Interestingly, prolonging mitosis using APC/C inhibitors can decrease the amount of mitotic defects in PSCs [26], suggesting that PSCs may not halt mitosis until all attachments are properly established.

## 3. Physiological Properties of PSCs Which Can Affect Mitotic Fidelity

Pluripotent stem cells possess distinct physiological characteristics that set them apart from differentiated cells, giving them unique properties that can influence the accuracy of cell division. These include differences in cell cycle length and checkpoint function, chromatin marks and organisation, mitotic machinery regulation and function and potential divergences in their tolerance and response to aneuploidy. In this part of the review, we will focus on these differences in physiology, consider how these physiological traits of PSCs can affect mitosis (Figure 2) and discuss potential strategies to overcome their suboptimal mitotic fidelity.

### 3.1. Cell Cycle

PSCs have atypical or compromised cell cycle checkpoints [58]. The cell cycle is comprised of discrete phases—mitosis, gap phase 1 (G1 phase), synthesis phase (S phase) and gap phase 2 (G2 phase). Each of these phases needs to be completed before cells enter the next phase, and to ensure this, checkpoint signalling pathways are activated if problems are detected. These checkpoints give cells the opportunity to correct any mistakes before progressing in the cell cycle [59]. For example, cells with entangled sister chromatids delay entry into mitosis to allow DNA topoisomerase II alpha to correct them. However, mouse ESCs complete cell division even in the presence of catenated sister chromatids, which results in aneuploid daughter cells [60]. Moreover, checkpoint activation can sometimes be uncoupled from the canonical response seen in differentiated cells, where, for instance, prolonged SAC activation leads to apoptosis during or after mitosis [61]. In contrast, SAC activation in PSCs is uncoupled from apoptosis [57], which may result in mitotic slippage in cells with improper attachments.

Pluripotent stem cells proliferate very fast, which may reflect the need for fast cell division in the embryo. They do this by shortening the G1 phase to approximately 2–3 h [38] (Figure 2). The short G1 phase is intrinsically linked to cell fate decisions, and G1 length increases upon differentiation [62,63]. Reciprocally, commitment of pluripotent stem cells to differentiation occurs during the G1 phase, and cells in the S and G2 phases are more impervious to exit the pluripotent state [64,65]. It is also during G1 that cells prepare for DNA replication by synthesising RNA, ATP nucleotides, amino acids and proteins and when components of centromeres and centrosomes—key structures for proper chromosome segregation—are replenished. CENP-A loading onto the centromeric chromatin occurs slowly during the long G1 of differentiated cells, for approximately 10 h [66]. It is therefore tempting to speculate that PSCs are not able to fully assemble CENP-A [24] in this short time window. The truncated pluripotent stem cell cycle can therefore lead to errors in mitosis by affecting the proper loading of CENP-A at the centromere or by not producing enough proteins and nucleotides for the next round of DNA replication.

### 3.2. Histone Post-Translational Modifications

Inducing pluripotency in terminally differentiated cells requires reprogramming, i.e., a genome-wide remodelling of the epigenome, such as histone modifications [67] and DNA methylation [68,69]. This remodelling allows for two cells with the same genome (iPSCs and the differentiated cells they were reprogrammed from) to have completely different cell identities and confers iPSCs with the characteristic plastic chromatin associated with the pluripotent state (Figure 2). This results in the repression of somatic genes and activation of self-renewal and pluripotency associated genes [70]. One of the earliest events in reprogramming is the rapid genome-wide re-distribution of H3K4me2 observed both in mouse and human iPSC reprogramming [71,72]. H3K4me2, together with H3K9me3, are considered barriers to reprogramming, as failure to remove or re-distribute these marks results in the inability of cells to reach pluripotency [67]. Moreover, PSCs have much lower levels of H4K20me1/2 when compared to differentiated cells [73]. Because chromatin structure and histone modifications also influence the recruitment of key mitotic players to chromosomes [74], it is possible that the lower levels and/or distinct distribution of specific histone modifications, such as H3K9me3, H3K4me2 and H4K20me1/2 in PSCs, influences the correct recruitment of pivotal mitotic players to mitotic chromosomes. Moreover, impaired recruitment of centromeric [75,76] as well as structural maintenance of chromosomes [77,78], kinases [79] and other proteins [78], to chromosomes, lead to defects in recruiting the downstream effectors of mitosis [80]. This characteristic chromatin of PSCs may be disrupting the fine-tuning of mitosis in these cells. Since this characteristically fluid chromatin is inherently linked to the pluripotent state, it is plausible that a compromised mitotic fidelity is an intrinsic characteristic of PSCs.

### 3.3. Mitotic Machinery of PSCs

Studies of the mitotic machinery in PSCs have, until recently, been mainly focused on the maintenance of stem cell identity [81]. In particular, cohesin has mainly been studied in the context of higher order chromatin organisation and the maintenance of stem cell identity [82], but it remains unknown how it functions in mitosis in these cells. Additionally, although condensation defects have been proposed as a cause for genomic instability in human PSCs [23], the underlying role of condensins, DNA topoisomerases as well as histone post-translational modifications important for chromosome condensation remain unknown. In addition to the structural components, critical regulators of chromosome attachments/checkpoint signalling have also been hypothesised to be altered in PSCs. The chromosomal passenger complex (CPC) component survivin was shown to be upregulated in ESCs [83]. Moreover, chemical inhibition of the critical CPC kinase Aurora B led to an increase in aberrant mitosis and apoptosis in ESCs [25]. How CPC is regulated in PSCs and how it affects mitotic fidelity is still unknown. The impact of the different regulation of all these proteins in mitosis and in the correct chromosome segregation of PSCs has not been investigated. Understanding which are the major differences in the composition, recruitment and regulation of the mitotic machinery of PSCs represents a clear knowledge gap. It is important to assess why these pathways are at least partially compromised and what are the mechanisms that become rate limiting during the extensive chromatin remodelling or upon the acquisition of pluripotency.

## 4. Consequences of Decreased Mitotic Fidelity in Pluripotent Stem Cells

The effects of aneuploidy have been studied in yeast, *Drosophila* and mammalian cancer and differentiated cells [9]. Aneuploidy can arise due to errors in mitosis, which can then impact the physiology of cells that inherit an abnormal chromosome number. Having an extra copy of a chromosome can lead to an imbalance in gene expression. Dosage compensation effects can overcome this problem, which is what happens in triple X syndrome (47, XXX). This is achieved by silencing the supernumerary X chromosomes by the dosage compensation mechanism of X chromosome inactivation [84]. Although in some species some autosomal dosage compensation similarly occurs [85], this has not been shown to naturally happen in mammalian systems so far. It has recently been shown that cancer aneuploid cells do not scale their protein expression to the DNA content [86,87,88], suggesting that some dosage compensation does occur (the mechanisms of which remain unknown). Even if DNA content and protein expression do not linearly scale, having extra copies of chromosomes results in increased expression of the genes of the trisomic chromosome, which results in extra protein production [89]. This can lead to several problems, since chaperones may become rate-limiting for correct protein folding and degradation, causing proteotoxic stress. Moreover, proper metabolism and replication need precise control of the exact stoichiometry of specific proteins/enzymes [85]. When cells produce extra amounts of proteins, the fine balance in these tightly regulated processes can be disrupted, resulting in metabolic and replication stress. Furthermore, having extra copies of chromosomes can in itself affect mitosis, resulting in chromosomal instability and in further mitotic stress [85,90]. Also, specific aneuploidies differentially impair the post-implantation developmental potential of human embryos, resulting in diverse developmental fates [91].

Karyotype analysis of several PSCs revealed that recurrent aneuploidies are commonly observed and proposed to directly impact on their biology (Figure 3). PSCs with chromosome 12 trisomy present higher tumorigenicity [15,92] and proliferate better than euploid PSCs in culture by increasing their replication potential [93]. Human PSCs trisomic for chromosome 12, 17 or both chromosomes 12 and 17, present higher incidence of mitotic errors and suffer increased replication stress than euploid PSCs [23]. Mouse stem cells trisomic for specific chromosomes were shown to have decreased differentiation potential and increased neoplastic properties [94]. Very recently, it was also shown that human PSCs trisomic for all chromosomes (triploid cells) have a decreased differentiation potential [95]. Some specific aneuploidies, such as those for chromosomes 1, 8, 12, 17 and X, are recurrent in PSCs [11]. The fact that there are recurrent aneuploidies also demonstrates that these cells are tolerant to specific aneuploidies, which may reflect that these chromosome gains confer advantages for cells that can then overtake euploid cells in culture. In differentiated cells, the stresses imposed by the aneuploid state ultimately lead to cell cycle arrest or cell death (Figure 3). In contrast, the high frequency of karyotypic abnormalities found in PSCs, including whole chromosome aneuploidies [10,11], implies continued proliferation despite the aneuploid state. This conundrum suggests that checkpoints are less functioning in PSCs. The intrinsic biological characteristics of pluripotency, such as increased proteasome activity, higher replicative stress and glycolytic-based metabolism, may affect the way they maintain cell homeostasis and tolerate aneuploidy. Studies in *Drosophila* have shown that aneuploid neuronal stem cells have delayed p53 activation, which confers tolerance to aneuploidy [96] and may impact on karyotypic evolution. It is possible that this also plays a role in human PSC tolerance to aneuploidy, as these cells frequently acquire and expand p53-dominant negative mutations [97]. However, the physiological pathways that render human PSCs tolerant to aneuploidy remain largely unknown. Uncovering this unique response to aneuploidy is crucial not only to understand critical aspects of stem cell physiology but also to allow for the elimination of these cells from PSCs cultures, similarly to what has recently been accomplished for cancer aneuploid cells [98].

## 5. Can We Increase the Mitotic Fidelity of PSCs?

Here we have summarized the recent evidence on the sub-optimal mitotic pathways of PSCs, which render these cells more prone to segregation defects and account for the elevated rates of aneuploidy. Characterising the mitotic machinery of PSCs opens the possibility of rescuing mitotic fidelity in these cells. Is it possible to modulate these pathways to obtain safer and more robust PSCs and accomplish their full potential?

It is possible that pluripotency cell identity is intrinsically linked with changes in mitotic machinery composition. For example, the intricate connection between cell cycle stages and cell fate [64] may impose an additional challenge on the attempts to overcome mitotic infidelity. It has been previously proposed that the acquisition of a unique pluripotent stem cell cycle organization during iPSC reprogramming is functionally linked to the acquisition of pluripotency. Consequently, modulating these mitotic pathways can prime cells to differentiate and consequently make them no longer pluripotent. Therefore, it also remains to be answered if restoration of mitotic fidelity is linked to loss of cell identity. This raises the hypothesis that aneuploidy is an intrinsic feature of pluripotency, implying that there is an intricate trade-off regarding sub-optimal mitosis and pluripotency.

On the other hand, attempts have been successful at decreasing mitotic errors in PSCs (e.g., prolonging mitotic duration or de-stabilization of kinetochore–microtubule attachments) [26]. Unfortunately, these revealed not to be a long term solution [26]. In contrast to what has been shown using similar strategies in fibroblasts [99], PSCs adapt to the drugs and stop responding to them, thus continuing to display errors in chromosome segregation [26]. This fast adaptation is similar to what was observed in cancer cells [100]. Nevertheless, it is possible that other strategies, manipulating other pathways and/or using other drugs may reveal more successful in the future.

## 6. Concluding Remarks

ESCs are derived from the inner cell mass of early embryos and can give rise to almost all the cell types of the organism. Induced pluripotent stem cell technology is a breakthrough methodology that relies on the ectopic expression of only four transcription factors to generate iPSCs [1]. PSCs, which include both ESCs and iPSCs, share many biological features and have an immeasurable potential to be used in research and in regenerative medicine. Yet, PSCs often display abnormal chromosome numbers (aneuploidy) which hinders their safe use. Mitosis is a carefully orchestrated and highly regulated process, which results in the equal segregation of the duplicated genome to the daughter cells. Understanding the mechanisms that support proper chromosome segregation in PSCs is an important step toward unravelling the underlying causes of compromised mitotic fidelity in these cells. Although recent studies are trying to delve into the mitotic machinery and mitotic fidelity of PSCs, we have still only unveiled the tip of the iceberg and several questions remain to be answered. By characterising the mechanisms regulating mitotic pathways in PSCs and/or conferring them increased tolerance to aneuploidy, we can design strategies to increase mitotic fidelity by modulating the key players of mitosis or selectively eliminating aneuploid cells from our cultures. Only then can we use these cells to their full potential.

## Figures and Tables

**Figure 1 ijms-24-11933-f001:**
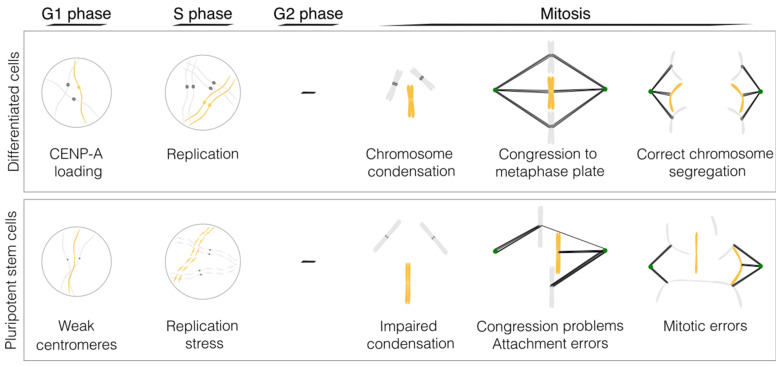
Molecular mechanisms underlying decreased mitotic fidelity in pluripotent stem cells. Schematic of the major mitosis problems in PSCs (please refer to Table 1 for the references and the main text for a detailed explanation). Figure designed in Inkscape v1.2.2.

**Figure 2 ijms-24-11933-f002:**
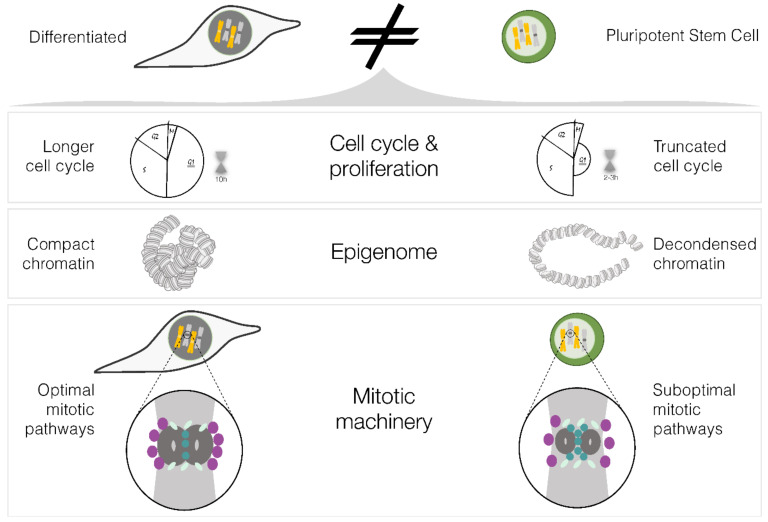
Distinct physiological characteristics of differentiated and pluripotent stem cells. Schematic of the physiological properties of pluripotent stem cells which can affect mitotic fidelity and cause the observed increased aneuploidy in PSCs (please refer to the main text for a detailed explanation). Figure designed in Inkscape v1.2.2.

**Figure 3 ijms-24-11933-f003:**
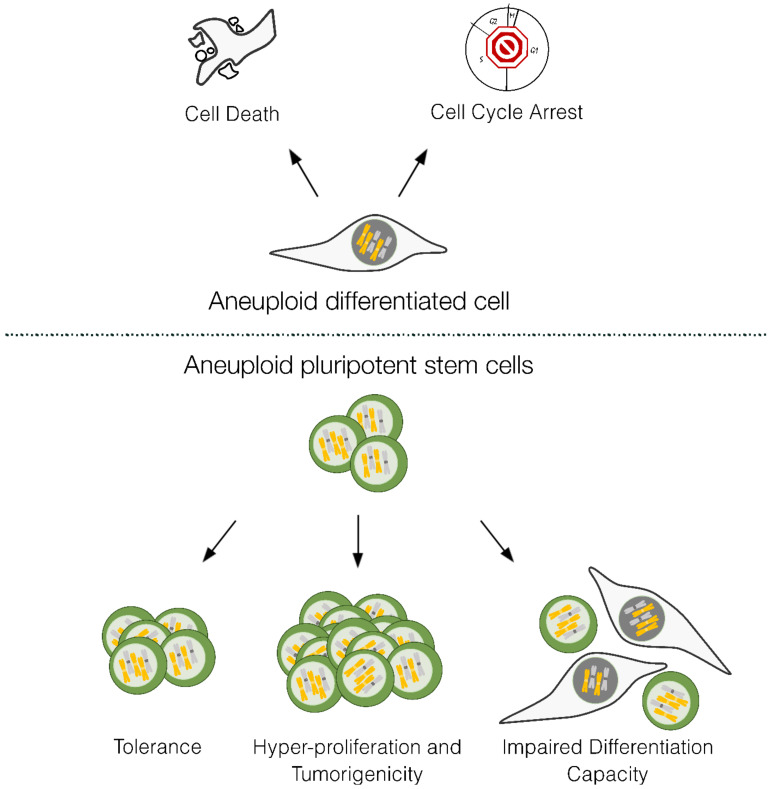
Response to aneuploidy in differentiated versus pluripotent stem cells. Differentiated aneuploid cells tend to activate cell cycle checkpoints which leads to arrest and can result in apoptosis. PSCs have higher tolerance to aneuploidy, can become hyperproliferative and have impaired differentiation capacity. Figure designed in Inkscape v1.2.2.

**Table 1 ijms-24-11933-t001:** Summary of the key references proving PSC have compromised mitosis and the cell types used in these studies.

Main Findings	Human ESCs	Human iPSCs
PSCs have recurrent chromosomal aberrations, including whole chromosome aneuploidies	[10,11,12,13,15,16]	[10,11,12,14,16]
PSCs have erroneous mitosis	[23,24,25,26]	[26,27]
PSCs undergo high replication stress during S phase	[23,27,28,29]	[23,27,28,29,30]
Aneuploid PSCs have problems in chromosome condensation	[23]	[23]
PSCs have weaker centromeres	[24]	[24]
PSCs have high number of erroneous kinetochore–microtubule attachments	[26,31]	-

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
