# Peer review of "Compromised Mitotic Fidelity in Human Pluripotent Stem Cells"

_ijms, 2023, doi:10.3390/ijms241511933_

Round 1
Reviewer 1 Report
Milagre and coworkers presented a very consistent review of the variables that impair the mitotic fidelity to Human Pluripotent Stem Cells, including the characterization of aneuploidy and the impacts of this failure on aspects of manipulation of this cell type. However, minor changes should be considered by the authors to improve the quality of the publication. As below:
A) Figures must include the program/software used to create them. Authors should also change the font of the letters used in Figure 2, which are difficult to read.
B) Authors may include a new figure illustrating the item "4. Consequences of decreased mitotic fidelity in pluripotent stem cells".
B) Authors may include a new figure illustrating the item "4. Consequences of decreased mitotic fidelity in pluripotent stem cells".
C) A point of attention is the generalization made about Embryonic Stem cells (ESC) and Induced-Pluripotent Stem Cells (iPSCs) both treated by PSCs. Despite the pluripotency similarity, the two cells are different, and the studies lead with one cell type or other cannot be extrapolated to the other cell, therefore, for this reviewer, the results discussed during the review must be shown as obtained from ESC or iPSCs and not generalized by PSCs. An alternative may be the inclusion of a Table that describes the most relevant results obtained from the articles that supported the review presenting the cell source (ESC or iPSC) for this result observed.
Author Response
Author's Reply to the Review Report (Reviewer 1).
We thank reviewer 1 for all the positive feedback. We have addressed all three points the reviewer raised, which we believe make the review more informative and clearer.
Reviewer comments A & B:
- A) Figures must include the program/software used to create them. Authors should also change the font of the letters used in Figure 2, which are difficult to read;
- B) Authors may include a new figure illustrating the item "4. Consequences of decreased mitotic fidelity in pluripotent stem cells".
Author’s reply to comments A & B:
We have now included the information regarding the program used in all figure legends. Moreover, to address point B, we included a new figure (Figure 3), which focus specifically on the consequences of decreased mitotic fidelity in pluripotent stem cells. In line with this new figure, we have now adjusted the original Figure 2 to restrict its focus on the physiological properties of PSCs. We also changed the lettering in figure 2, which, for unknown reasons, got unformatted in the reviewer’s version of the paper.
Reviewer comment C:
- C) A point of attention is the generalization made about Embryonic Stem cells (ESC) and Induced-Pluripotent Stem Cells (iPSCs) both treated by PSCs. Despite the pluripotency similarity, the two cells are different, and the studies lead with one cell type or other cannot be extrapolated to the other cell, therefore, for this reviewer, the results discussed during the review must be shown as obtained from ESC or iPSCs and not generalized by PSCs. An alternative may be the inclusion of a Table that describes the most relevant results obtained from the articles that supported the review presenting the cell source (ESC or iPSC) for this result observed.
Author’s reply to comment C:
To address the reviewer’s concern, we have now included a table which discriminates mitotic defects observed according to the cell type and the respective references.
Reviewer 2 Report
This manuscript highlights the molecular mechanisms that impair mitotic fidelity in human PSCs. The overall outline of this article is good and well-written. Below are a few comments that need to be addressed:
1. There are a few missing references, mainly when a statement is inferred from several papers on lines 55, 61,103, 112, and 117.
2. Explain which culture conditions affect the genetic stability of PSCs on line 55.
3. The sentence on lines 101-103, “PSCs that do not activate 101 apoptotic pathways in S phase due to DNA replication problems…..” is complex and needs to be split into two sentences or reworded
4. Line 115 is ambiguous and should be rephrased.
5. On line 148, to what extent does the weak centromere affect the recruitment of kinetochore proteins in mitosis
5. The font quality and size in Figure 2 could be improved. Some letters are overlapping, e.g., Epigenome".
6. A table summarizing this review will be helpful for readers.
7. Line 61-63 explains the focus of this review. This could be explained further to emphasize the strengths of this review.
8. This review could include more figures explaining the different mechanisms.
The quality of English is good but could be improved. A few grammatical errors and complex sentences have to be corrected.
Author Response
Author's Reply to the Review Report (Reviewer 2)
We thank reviewer 2 for the positive feedback regarding our review. We have addressed the points raised by reviewer 2. We believe the review is now more comprehensive and easier to read.
Reviewer’ comment 1: There are a few missing references, mainly when a statement is inferred from several papers on lines 55, 61,103, 112, and 117.
Author’s reply:
We have added new references to strengthen the statements the reviewer indicated in lines 55 and 117.
The sentence in line 103 has been rephrased and appropriate references included.
The sentences in lines 61 and 112 are meant to describe the subsequent focus on the review. Hence, we opted for not including them in the general sentence so that each finding and the corresponding reference could be presented in more detail.
Reviewer’ comment 2: Explain which culture conditions affect the genetic stability of PSCs on line 55.
Author’s reply: We have clarified the culture conditions that have been shown to affect the genetic stability of PSCs: “such as medium acidification” has been added to the sentence.
Reviewer’ comment 3. The sentence on lines 101-103, “PSCs that do not activate apoptotic pathways in S phase due to DNA replication problems…..” is complex and needs to be split into two sentences or reworded.
Author’s reply: This sentence has been rephrased to improve clarity: “Although apoptosis is PSCs’ preferred response to DNA damage [38], they can proceed to mitosis after replication stress [31,35]. This can result in either abortive or erroneous mitosis, which can then result in daughter cells that are aneuploid.”
Reviewer’ comment 4. Line 115 is ambiguous and should be rephrased.
Author’s reply: We have changed the beginning of the paragraph as follows:“In interphase, PSCs chromatin is more open, mobile and accessible [30,40] than that of differentiated cells and extensive remodelling of the chromatin during iPSC reprogramming is necessary to induce pluripotency [41]. It is conceivable that these differences in mitotic chromatin properties affect how the mitotic machinery functions and is regulated in these cells, which can impact efficient chromosome segregation.”
Reviewer’ comment 5. On line 148, to what extent does the weak centromere affect the recruitment of kinetochore proteins in mitosis
Author’s reply: We have now added “resulting in reduced levels of key proteins in kinetochore-microtubule attachment and chromosome motility” to emphasize the reported decrease in some kinetochore proteins.
Reviewer’ comment 5. The font quality and size in Figure 2 could be improved. Some letters are overlapping, e.g., Epigenome".
Author’s reply: The figure included in the pdf appeared with formatting issues for unknown reasons. We have now uploaded a new version (with slight modifications to address reviewers’ points).
Reviewer’ comment 6. A table summarizing this review will be helpful for readers.
Author’s reply: We have now included a new table summarizing the key findings discussed in this review with regard to described mitotic pathways impaired in PSCs. The other points discussed in the manuscript (e.g. properties of stem cells or differences in the consequences to aneuploidy) are less well established and therefore are only presented in the figures.
Reviewer’ comment 7. Line 61-63 explains the focus of this review. This could be explained further to emphasize the strengths of this review.
Author’s reply: We have extended this paragraph to outline the topics discussed in the present review. We have also included a graphical abstract with a similar summary.
Reviewer’ comment 8. This review could include more figures explaining the different mechanisms.
Author’s reply: We have now included a new figure (Fig. 3) focusing on the consequences of aneuploidy in stem cells, which was the section not yet outlined as a figure.

Round 2
Reviewer 1 Report
The authors presented a revised version of the manuscript with significant improvements according to the reviewer's suggestions.